# Global Trends in Cancer Nanotechnology: A Qualitative Scientific Mapping Using Content-Based and Bibliometric Features for Machine Learning Text Classification

**DOI:** 10.3390/cancers13174417

**Published:** 2021-09-01

**Authors:** Nuwan Indika Millagaha Gedara, Xuan Xu, Robert DeLong, Santosh Aryal, Majid Jaberi-Douraki

**Affiliations:** 11DATA Consortium, Kansas State University Olathe, Olathe, KS 66061, USA; mgnindika@ksu.edu (N.I.M.G.); xuanxu@ksu.edu (X.X.); 2Department of Mathematics, K-State, 22201 W Innovation Dr. Olathe, Olathe, KS 66061, USA; 3Nanotechnology Innovation Center Kansas State, Department of Anatomy and Physiology, College of Veterinary Medicine, Kansas State University, Manhattan, KS 66506, USA; robertdelong@vet.k-state.edu; 4Department of Pharmaceutical Sciences and Health Outcomes, The Ben and Maytee Fisch College of Pharmacy, The University of Texas, Tyler, TX 75799, USA; saryal@ksu.edu

**Keywords:** cancer, nanotechnology, nanomaterials, bibliometric measures, machine learning models, visualizing networks

## Abstract

**Simple Summary:**

This study is a new way of providing potential opportunities for prevention, diagnosis, and therapy to investigate the comprehensive trends in cancer nanotechnology research. This paper applied the qualitative method of bibliometric analysis on cancer nanotechnology using the PubMed database during the years 2000–2021. It mined nearly 50,000 papers published in multiple reputed journals. The impact of our findings is significant, which focuses on hybrid medical models and content-based and bibliometric features for machine learning models in cancer detection, diagnosis, imaging, and therapy related to cancer nanotechnology in the world. We mainly identified and classified the top and significant keywords, countries, authors, affiliations, and research areas representing the documents in the top 100 journals in cancer nanotechnology, which will help researchers explore more powerful anticancer nanomedicines in the next five to ten years.

**Abstract:**

This study presents a new way to investigate comprehensive trends in cancer nanotechnology research in different countries, institutions, and journals providing critical insights to prevention, diagnosis, and therapy. This paper applied the qualitative method of bibliometric analysis on cancer nanotechnology using the PubMed database during the years 2000–2021. Inspired by hybrid medical models and content-based and bibliometric features for machine learning models, our results show cancer nanotechnology studies have expanded exponentially since 2010. The highest production of articles in cancer nanotechnology is mainly from US institutions, with several countries, notably the USA, China, the UK, India, and Iran as concentrated focal points as centers of cancer nanotechnology research, especially in the last five years. The analysis shows the greatest overlap between nanotechnology and DNA, RNA, iron oxide or mesoporous silica, breast cancer, and cancer diagnosis and cancer treatment. Moreover, more than 50% of the information related to the keywords, authors, institutions, journals, and countries are considerably investigated in the form of publications from the top 100 journals. This study has the potential to provide past and current lines of research that can unmask comprehensive trends in cancer nanotechnology, key research topics, or the most productive countries and authors in the field.

## 1. Introduction

Nanomaterials are perhaps the most important scientific advancement in the last decade and have revolutionized many segments of society and technology including computers and electronics, engineering, military applications, and many others. There is no more important application benefitting human health than nanomedicine, indeed cancer nanotechnology seeks tfo apply nanoparticles and nanoconstructs to improve cancer detection, diagnosis, imaging, and therapy while reducing toxicity associated with traditional cancer therapy [1,2]. A great deal of information in this important new cancer nanotechnology emerging sub-discipline has been published. Thus, to inform the field and provide guidance to researchers, clinical practitioners, and nanotechnologists, it is important to take stock of where the field stands today, in order to see the opportunities and challenges for the future.

Numerous topics related to the applications of cancer nanotechnology were studied, from cancer detection and diagnosis to tumor imaging, drug delivery, and cancer therapy, and mainly concerned with the development in nanotechnology for the future of clinical cancer care. Our aim in this study was to collate and organize this wealth of information to investigate global directions and trends of cancer nanotechnology research from appropriate datasets of accredited literature, independent hubs, and scholarly research sources. We accumulated all data on cancer nanotechnology from the PubMed database during 2000–2021 [3]. This analysis shows what direction the field has previously been going and is currently trending toward, and how the field has changed by exploring the most notable countries, common keywords, authors, institutions, and journals.

Great advancements in cancer nanotechnology have come in drug delivery, development of new materials, and a basic understanding of nanoparticle pharmacokinetics, biodistribution, and biological and clinical activity [4,5,6,7], one major direction being “monitoring, repair, and improvement of human biologic systems” [8]. The link of cancer nanotechnology into clinical practice requires careful clinical, ethical, and societal consideration and a multidisciplinary approach. Advances in combination therapies based on transdisciplinary approaches have been made possible by interconnecting technology developers, physicists, chemists, and data scientists collaborating with clinicians and biologists to identify and devote effort to principal complications and enigmas, and clinical translation of cancer care and treatment [9,10,11,12,13,14,15,16]. Multiple studies have shown that cancer nanotechnology has significant potential to improve current standards of care [17,18,19]. In addition, a variety of nanomaterials were under investigation and development with the applications related to cancer nanotechnology, including biodegradable controlled-release polymers and polymeric nanoparticles, the dendrimer-mediated formation of multicomponent nanomaterials (e.g., receptor-targeted/peptide-conjugated dendrimer-encapsulated nanoparticles), lipid-based microparticles, organometallic complexes, and carbon- and silicon-based nanostructural materials [2,17,20,21,22,23]. Biological performance of materials, biocompatibility, safety and toxicology of engineered nanomaterials, size distribution and size-dependent diffusion, surface chemistry, and their properties in biologic systems are also considered in the selection of specific nanomaterials for applications in cancer nanotechnology. On the other hand, Rueda G. et al., investigated the nanotechnology field using bibliometrics and social network analysis in 1992–2006 [24]. They examined the inter-relationships among lead authors and co-authors, authors with the highest number of publications, and countries making the highest contributions to nanotechnology.

We implement the qualitative method of text-based and content-based classification called bibliometric analysis on cancer nanotechnology. Bibliometric-enhanced information retrieval is a systemic meta-study evaluating research performance with data from multiple publications and citation resources for text mining and machine learning models. The bibliometric analysis of the existing research is an important tool to investigate scientific research developed on different topics. Bibliometrics impacts the progress of science in different ways: for example, by allowing assessment of progress made, identifying the most trustworthy sources of scientific publications, laying the academic foundation for assessing new developments, and identifying major scientific actors and content-based features [25]. Open Knowledge Maps is another tool that helps to visualize the research findings for science and is a better way to explore and discover scientific research papers for bibliometric analysis [26].

Thus, this paper carries out a thorough bibliometric analysis of cancer nanotechnology applications based on all the available publications throughout the past 21 years, which allows new researchers to learn how the fields are being explored and evolved in cancer nanotechnology. For this purpose, descriptive statistics analysis as an important part of machine learning was put into effect to quantitatively characterize and outline features of collected data, and semantic mapping analysis for multiscale data structure, and network analysis to represent and visualize data are used in this analysis. The purpose of this study from a multifaceted approach is to: (1) distinguish major words in abstracts, including keywords and their evolution, to determine and represent magnitude and direction of the field of study; (2) visualize clusters of scientific collaborations among authors and affiliations, and authors’ collaborative efforts from different countries; (3) identify productive publication countries, journals, authors, and affiliations in the cancer nanotechnology research field; and (4) explore and identify research areas under nanotechnology and the top cancer types. To the extent of our knowledge, which relies on the cancer nanotechnology database of over 50,000 publications we curated from the 2000–2021 PubMed database, no bibliometric analysis has been conducted in the field of cancer nanotechnology. Therefore, this study could provide us with original findings and important information, and insight into cancer nanotechnology’s dynamics and direction.

In summary, this study analyzes a total of 48,629 articles that 166,672 authors published on the cancer nanotechnology theme in 1701 journals, and they are identified for the analysis of global scientific production during the period ranging from 2000 to 2021 related to cancer nanotechnology using the PubMed database. Using this dataset, we further divided the documents into two samples: documents published in the top 100 or 50 journals using the journal impact factor (IF) as a scientometric index calculated by Clarivate [27] ranging from 5 ≤ IF ≤ 245 or 8 ≤ IF ≤ 245, respectively. The author’s keywords in this analysis are classified into different clusters based on the samples. This showed that the studies focused on the research of nanotechnology, nanoparticles, and cancer are the most used topics in the area of cancer nanotechnology. We found that the USA and China are the most productive countries in cancer nanotechnology, followed by the UK and India. In addition, the USA institutions have appeared on the list of most productive institutions in terms of publications, with the University of California among the highest. It was clear from the bibliometric analysis that the *International Journal of Nanomedicine*, and *ACS Applied Materials and Interfaces* are the journals with the most frequently cited papers. Cell lines, cancers, nanoparticles, detection, and therapy are some of the most frequently used co-occurring keywords among the samples, while breast cancer, lung cancer, prostate cancer, and colon cancer are among the top cancer types. Furthermore, drug delivery and delivery systems, cancer therapy, DNA nanotechnology, RNA nanotechnology, breast cancer, and drug resistance are among the top and significant research areas in nanotechnology.

## 2. Materials and Methods

We used the PubMed search engine to collect documents about “cancer nanotechnology”. PubMed is considered a free search engine that contains the MEDLINE database of abstracts and references on biomedical topics and life science. Using the search query key “(nano* AND (carcinoma OR sarcoma OR blastoma OR tumor OR melanoma OR glioblastoma OR leukemia OR lymphoma OR cancer)”, we collected documents ranging from the first quarter of 2000 (Q1) to the first quarter of 2021 (Q1). The query key “nano*” with the asterisks at the end is a special case that uses a regular expression and identifies all the possible nano types, extending to almost 800 different nano-related terms. Here are the top 20 identified nano-related keys: nanoparticle, nanotechnology, nanomedicine, nanomaterial, nanocarrier, nano, nanoscale, nanostructure, nanotube, nanocomposite, nanorod, nanosystem, nanogel, nanoformulation, nanoplatform, nanoprobe, nanocapsule, nanofiber, and nanodevice. The search was also made for all possible variations of a word that are presented by different authors, for instance, the term “nanoparticle” represents the combination of “nanoparticle(s)”, “nano-particle(s)”, or “nano particle(s)”.

We were also able to retrieve a total of 52,083 documents. Key elements, including PMID, title, authors, affiliations, abstract, keywords, published year, and the journal were collected for each document. A developed Python 3.7.3 and R version 3.6.2 program was applied to extract key elements and the number of citations. We used the PubMed search engine to find the number of citations for each document. The above-mentioned key elements with all the documents between 2000 (Q1) and 2021 (Q1) were considered as datasets and performed to obtain the analysis with the goal of obtaining a general vision of the field. Compared to all publications, we used the journal-level metric IF as a scientometric index introduced by the Institute for Scientific Information and currently published by Clarivate [27] to further divide the dataset into the top 100 journals (5 ≤ IF ≤ 245) and top 50 journals (8 ≤ IF ≤ 245) with respect to high-quality/high-impact publications. It is worth mentioning that there are a number of publications indexes and factors regarding measuring scientific activities and publication impacts, including articles citations, h-index, i10-index, SJR Q indexing, etc. [28,29], but we eventually decided to utilize the journal-level metric IF, which may represent the better measure as this journal-level metric is globally accepted. These stratifications help us analyze and differentiate high-quality/high-impact publications as opposed to all the publications combined. Therefore, we analyzed three samples and made a comparative analysis to identify how significantly high-quality/high-impact publications change the course and field of cancer nanotechnology.

In this study, we also used the descriptive statistics method to obtain the distribution and summary statistics of the dataset, including publication distribution by year, countries, journals, authors, and affiliations. Descriptive statistics optimally simplify large amounts of data by presenting quantitative descriptions along with simple graphics analysis.

The R programming language and QGIS version 3.10 (QGIS Development Team) allow us to visualize the summary statistics and transient patterns in the dataset. According to the author’s address information, the corresponding affiliations and countries were manually preprocessed and identified with an automated process. We applied four types of bibliometric techniques. First, we used Geomap and heatmap, which visualize the publications by country over the studied period. Second, we performed a co-word analysis to establish relationships between documents through keyword co-occurrences. VOSviewer version 1.6.0 software (Leiden University, Leiden, the Netherlands: https://www.vosviewer.com/, accessed on 15 August 2021) was used to extract key terms (including single words and phrases) based on authors’ keywords fields.

Further, we investigated and visualized co-occurring cancer types and nanoparticles in the area of cancer nanotechnology. The third was co-author analysis, which investigates the relationship between leading authors and the most cited references through the map and clusters visualization. The fourth was co-citation analysis, which provides and visualizes the top-cited journal co-occurrences, institutions, and the impact factor. IFs of the journals were analyzed using the R program package ‘scholar’. We used VOSviewer software to analyze the association between the most productive authors and the most cited references to generate the network map and clusters visualization. Finally, we augmented qualitative text analysis using natural language processing (NLP) to investigate key characteristics and the pattern associations with nanotechnology and the top 10 cancer types [30,31]. Under the qualitative method, we mainly focus on qualitative text analysis or qualitative content analysis (e.g., thematic analysis). Qualitative text analysis requires researchers to read data, assigning code labels as succinct descriptors of meaning to text segments, and iteratively developing findings [32,33,34,35]. However, qualitative text analysis is laborious and resource-intensive as researchers seek an in-depth understanding of large text-based data [32], in our case approximately 50,000 records of publications. Thus, researchers are usually limited to smaller sample sizes when analyzing text-based data. One potential approach to address this concern is NLP. NLP has the potential to automate part of this qualitative process. Further, NLP can analyze unlimited amounts of text-based data in scientific literature where most of the data is in an unstructured format without fatigue and in a consistent, unbiased manner. In this research, we have nearly 50,000 documents; thus, we augmented qualitative text analysis using NLP to investigate key characteristics and the pattern associations with nanotechnology and cancer types.

## 3. Results

The design of our study was both rigorous and comprehensive. The literature search retrieved 52,073 records during the period ranging from 2000 (Q1) to 2021 (Q1), of which 48,629 articles’ records were included in the final analysis (Figure 1). From the total search results, we excluded non-English studies (n = 2214), empty abstract (n = 318), and 31 systematic reviews (n = 31). Thus, we obtained a total of 48,629 documents that were published by 188,676 unique authors (from a total of 381,752 non-unique authors as each author can have more than one publication) on the theme “cancer nanotechnology” in 1701 journals (Table 1). Figure 2 shows the number of documents that were published per year. It clearly shows that the number of publications increased over time at an average rate of 1000 documents per year (Figure 2). A notable escalation in the overall publication rate is evident from 2010. These numbers are also given for the top 100 journals and top 50 journals correspondingly in the following columns of Table 1 and Figure 2.

Figure 3a–c represents the snapshot of heatmaps of the countries’ publications in the area of cancer nanotechnology from the PubMed Core Collection in 2021 for the entire dataset (considered all 48,629 documents), the sample contains the top 100 journals and the top 50 journals. In 2020, the USA and China were the most productive countries in terms of publications in the field of cancer nanotechnology and, within the past decade period of 2010–2021, the number of publications spread in most countries in Europe, South Asia, and East Asia (Figure 3a). The same analysis was equally performed for the top 100 and 50 journals. Figure 3b indicates the sample of the top 100 journals averagely covering the publication of most of the destinations in the world in 2020, while a sample of the top 50 journals only covers the countries such as China, the UK, and some parts of the USA and Europe. This leads to the idea of high-quality/high-impact publications in sub-samples that contain the top 100 journals, significantly impact the cancer nanotechnology field, and also cover an average of 50% of publications. Figure 3d is the Geomap of each country’s publications, which visualizes the map of each country, with colors and values assigned. For more information and better quality, readers are encouraged to check Appendix A for heatmaps of the number of publications in cancer nanotechnology between 2000 to 2021 in the US, Europe, and Asia.

During the entire publication period, the most productive countries by authors and documents were China (120,431), the USA (85,215), and the UK (17,434), followed by India (14,680) and South Korea (11,975) (see Figure 4).

The network of co-occurring authors’ keywords is presented in Figure 5 for the three scenarios. The analysis of keywords reveals a high heterogeneity of terms within different samples; the top 100 keywords are used >10 times in authors’ keywords of all documents and are illustrated in the right-hand side panels of Figure 5. The keywords list typically reflects one of the main focuses of a paper. In the keywords, we can unsurprisingly observe that “nanotechnology” is an important concept in the published documents. The most frequently used co-occurring keywords in the entire dataset are as follows: Therapy/Drug/Treatment, cells, cancer, nanoparticles, delivery, imaging, in vivo, or detection. These keywords are also represented by other samples of the top 100 or 50 journals heterogeneously (see Table 2). The analysis re-organized all keywords and grouped similar terms to offer a broad picture of cancer nanotechnology.

We also identified some of the top keywords such as cells, cancer, delivery, nanoparticles, imaging, and therapy that are also in the sample of the top 100 journals (Table 2). Figure 6a,b gives the top 10 nano-related keywords and cancer types, respectively. In terms of the top 10 nano-related keywords, nanoparticles (63,011), nanocarriers (6124), nanomaterial (4556), nanomedicine (4271), nanotechnology (3163), and nanotubes (2925) are the most frequently used keywords. When turning to the top 10 cancer types, it is interesting to report that the analysis is focused on breast cancer (13,114), lung cancer (4962), prostate cancer (3674), colon cancer (3451), ovarian cancer (2439), and pancreatic cancer (2269) among all cancer types. We can unsurprisingly observe that approximately 50% of these nano types and cancers are present in the sample of the top 100 journals and 25% in the top 50 journals.

The top 10 journals that represent each sample, including the impact factor of publishing in the area of cancer nanotechnology, are indicated in Table 3. The top 10 journals published 12,657 papers in the field of cancer nanotechnology comprised 26% of the total. The International Journal of Nanomedicine (IF 4.471; 2018) had the most significant number of publications with 2149 articles, followed by ACS Applied Materials and Interfaces (IF 8.33; 2018) with 1839 papers and Biomaterials (IF 10.27; 2018) with 1790 documents. From Table 3, we note that the journal ACS Applied Materials and Interfaces, Biomaterials, ACS Nano, Biosensors and Bioelectronics, and Nanoscale are high-quality/high-impact publications and widely collect the publications made in the subject of cancer nanotechnology. It clearly indicates 20% of total publications are presented by the sample of the top 100 journals, which range from an impact factor greater than 5 and less than 244.

Figure 7a shows the resulting co-author network and the density map. The top 10 most productive authors had a total of 1716 papers. Chen, Xiaoyuan at the National University of Singapore (Singapore), is the author with the highest number of publications (259) in the area, followed by Liu, Yang at the Chinese Academy of Sciences (China), who produced 201 articles. Wang, Wei, at Brigham and Women’s Hospital, Boston, Massachusetts (USA), Robert S. Langer, at MIT (USA), Kim, Kwangmeyung at Korea Institute of Science and Technology (KIST) (South Korea), and Leaf, Huang, at the University of North Carolina at Chapel Hill (USA), published more than 100 papers. Farokhzad, Omid at Harvard Medical School (USA), and Atyabi, Fatemeh at Tehran University of Medical Sciences (Iran) equally published 80 articles in the area. Moreover, in the sample of the top 100 journals, the top 10 most productive authors have presented 40% of the articles from the entire dataset. For more information, the density map also shows the concentration of the co-author network (Figure 7b).

Figure 8 shows the top 10 institutions which present the organizations with the highest production of articles in cancer nanotechnology by the top 10 most published countries. Most documents are mainly from the University of California, USA, and Shenyang Pharmaceutical University, China, followed by institutions in Iran, Korea, Japan, and the UK. On average, more than 40% of documents are covered by institutions in the sample of the top 100 journals.

We further investigated the research areas in nanotechnology and the top 10 cancer types (see Figure 9a–c and Figure 10a–c). The Figure 9a chord diagram represents the connections between research areas and nano-related terms with different colored segments. The thickness of the ribbon is proportional to the significance of the flow: drug delivery and delivery system, cancer cells, iron oxide, carbon nanotube, side effect, silica, and gold nanoparticles are the top and significant research areas in nanotechnology. We identified most of these top and significant keywords in the sample of the top 100 and 50 journals (See Figure 9b–c). The connection between the research areas and cancer types is displayed in a circular layout regarding the top cancer types in three different scenarios (Figure 10a–c). For example, topics in prostate cancer include PC3 cells, iron oxide, xenograft model, stem cells, and selenium nanoparticle, as some of the areas observed within the segment. Similarly, these research areas among the top 10 cancer types in the entire dataset sample were also presented by the top 100 and 50 journals in Figure 10b,c, respectively. Based on Figure 10a–c, we further investigated the overlapping research areas in the top 10 cancer types in Figure 11. Carcinoma cells, stem cells, xenograft model, delivery system, cytotoxic effect, cancer therapy, and iron oxide are some of the areas that overlap highly among the top 10 cancer types.

## 4. Discussion

This paper is a first-of-its-kind investigation in the area of cancer nanotechnology providing potential opportunities for prevention, diagnosis, and therapy to study the comprehensive trends in cancer nanotechnology research. In this study, we analyzed the global scientific production from the period ranging from 2000 to 2021 related to cancer nanotechnology. Our results showed an increase in the cumulative volume of documents worldwide and a tendency to continue growing in terms of publication numbers. Based on our findings, we can conclude that the USA and China are the most productive countries in the field of cancer nanotechnology, followed by the UK, India, Korea, and Iran. Based on the availability of resources among countries, excellent research emerges in cancer nanotechnology, such as that in Northern Europe, Iran, and India. Among European countries, the study confirms the UK ranking first in the quantity of scientific production. Large countries, such as the UK, Italy, Germany, and France, published the highest number of papers. Among non-EU countries (besides China, with the highest numbers of published articles (120,431)), scientists from India (14,680), and South Korea (11,975) are the top publishers and researchers. It is worth mentioning that nation rankings changed considerably when other conditions were considered, such as the primary affiliations of authors and co-authorship networks in the area of cancer nanotechnology. 

During the first five years of observation, the number of papers in the area of nanotechnology was very low. However, after 2010, the publications steadily increased all over the world, but despite this final discrepancy, the USA, China, and the UK have increased their production over time. The overall production increased by 211% comparing 2010 to 2021.

The highest production of articles on cancer nanotechnology is mainly from the USA institutions. The University of California, Shandong Pharmaceutical University of China, University College of London, CSIR-Indian Institute of Chemical Technology and Amirata Institute of Medical Sciences, India, and Seoul National University, South Korea, have published the largest number of articles. There were a number of highly cited authors including, not surprisingly, Chen, Xiaoyuan at the National University of Singapore (Singapore); Liu, Yang at the Chinese Academy of Sciences (China); Wang, Wei, at Brigham and Women’s Hospital, Boston, Massachusetts (USA); Kwangmeyung, Kim, at Korea Institute of Science and Technology (KIST) (South Korea); Leaf, Huang, at the University of North Carolina at Chapel Hill (USA); Farokhzad, Omid at Harvard Medical School (USA); and Atyabi, Fatemeh at Tehran University of Medical Sciences (Iran).

It is also useful to mention that the International Journal of Nanomedicine, and ACS Applied Materials and Interfaces, and ACS Nano are the journals with the most frequently cited papers. We found that the top journals in the entire dataset are: International Journal of Nanomedicine *>* ACS Applied Materials and Interfaces *>* Biomaterials > Nanoscale > ACS Nano > International Journal of Pharmaceutics > Scientific Reports ≈ Analytical Chemistry. However, the story is different when focusing on the top 100 or 50 journals. We observed that the top journals with the most frequently cited papers are obtained from, among others, ACS Applied Materials and Interfaces, Biomaterials, Nanoscale, ACS Nano, Nat Nanotechnology, or Nano Letter. We also note that the journals such as *ACS* Applied Materials and Interfaces, Biomaterials, ACS Nano, Biosensors and Bioelectronics, and Nanoscale are high-quality/high-impact publications widely collecting papers in the subject of cancer nanotechnology.

Additionally, the query key nanoparticle has widely collected publications, but nanotechnology was also within the top 10 nano-related keywords. There are a wide number of research topics that may interest scientists to investigate in the future but are not limited to the treatment strategy of metastatic cancer using nanotechnology, for example, breast cancer. Similarly, drug delivery, DNA and RNA, therapeutic efficacy, radiation therapy, detection, and tumor are the research interest topics dedicated to nano-related keywords. Additionally, carcinoma cells, stem cells, xenograft model, delivery system, cytotoxic effect, cancer therapy, and iron oxide are some of the research areas that overlap among the top 10 cancer types. During this search, we also found a notable interest in the research community on biomimetic nanotechnology in which synthetic and biologics such as cell extracellular vesicles have been exploited as drug delivery solutions. Thus, all of these topics are essentially related to the applications of cancer nanotechnology. We mainly identified more than 50% of information related to the keywords, authors, institutions, journals, and countries, which are significantly presented in the top 100 journals. Additionally, we found there is no significant difference in information between the documents in the top 100 journals vs. the top 50 journals. Further, this study shows that cancer nanotechnology can improve a large number of scientific applications in society.

When this project was started, the focus of our work was to perform research topics specifically related to cancer nanotechnology. For this purpose, finding a perfect measure to further divide the documents based on keywords and main themes was not a straightforward task and that is the main reason we selected an internationally recognized measure, IF, to index and parse all the publication records. The advantage of using this journal-level metric was that we were able to investigate three datasets (entire dataset, top 100 journals, and top 50 Journals) and make a comparative analysis to identify how significantly high-quality/high-impact publications change the course and field of cancer nanotechnology. It was clear that the entire dataset containing almost 50,000 published documents provides all field-related information about cancer nanotechnology. Using IF ≥ 5, we were then able to implement our data-mining techniques on papers from the top 100 journals which, roughly speaking, covers almost all journals in the field of nanotechnology. Finally, it is reasonable to say that the benefit of using IF ≥ 8 was that it could potentially include all the top journals that disseminate manuscripts in nano-related fields. Using this strategy, we could clearly see global patterns and changes when we applied this formula to our three datasets, for instance, see Table 2 and Table 3 or Figure 9 and Figure 10.

Another important reason that we could not make use of a different journal-level metric other than IF was that the journal’s aims and scopes usually cover a broad range of topics which are typically inconclusive to select a very specific field of research. We understand that it is necessary and imperative for them to cast a wide net of research topics to bring together extensive research relevant to the journal’s audience, as one of their main goals is to target a large, general readership. For this reason, we believed it might have been a convoluted task to identify measures other than IF to filter for different datasets. Furthermore, we had not had a priori knowledge to do predefined filtering based on different topics or cancer types, or nano-related materials. It first necessitated to analyze the data and then see the trends in each field to comprehend whether it was required to identify any other measures or not. Now that different patterns can be observed using this measure and the techniques used in this study, it would be noteworthy to further mine the data for other types of patterns using these key research topics for future work. As previously mentioned, one limitation we may anticipate is that identifying a specific field might involve arduous labor to distinguish relevant journals and that journals cover a wide range of research topics in their scopes and aims. 

## 5. Conclusions

Cancer nanotechnology has globalized over the last 10 to 15 years, with a few papers beginning around 2001–2002 to more than 48,000 articles as of May 2021. The interest in this field has expanded exponentially the curve fit of publications, suggesting more than 6000 publications in 2020, with even more records predicted based on the curve trajectory for the next decade of 2020–2030. The heatmap and Geomap suggest that the field was incubated initially in the technology hotspots in the US in the early 2000s in the Silicon Valley Bay Area, Boston, Chicago, and North Carolina Research Triangle area, the UK, Europe, and China. By 2015 there was a clear expansion throughout much of the US and across most of Europe and Asia. By 2020, cancer nanotechnology has clearly become a global science with activity on every continent. We selected the top 10 keywords in the entire dataset and compared those with the other two different samples, and the top keywords among these samples are cells, cancer, and nanoparticles. Top institutions publishing cancer nanotechnology work included: University of California, USA; Shandong Pharmaceutical University, China; University College of London; CSIR-Indian Institute of Chemical Technology and Amirata Institute of Medical Sciences, India; and Seoul National University, South Korea. In terms of sub-disciplines or sub-categories, the top cancer type studies by more than two-fold were breast cancer, followed by lung, prostate, colon, ovarian, and pancreatic, in that order. Chord plot analysis showed the greatest overlap between nano-related keywords and DNA, RNA, mesoporous silica, breast cancer, cancer diagnosis, and cancer treatment. Circos plot analysis showed multiple pattern associations with nanotechnology, not only for cancer and nanoparticle types but also cancer cell lines and biomarkers, mouse models, and various techniques. Overall, the data combined reflect an ever-increasing international research effort in cancer nanotechnology. With cancer being a leading cause of human mortality and suffering, the hope is that these early research efforts will now begin to pay off in translation through preclinical animal models and the clinic to more specific and more powerful anticancer nanomedicines in the next five to ten years.

## Figures and Tables

**Figure 1 cancers-13-04417-f001:**
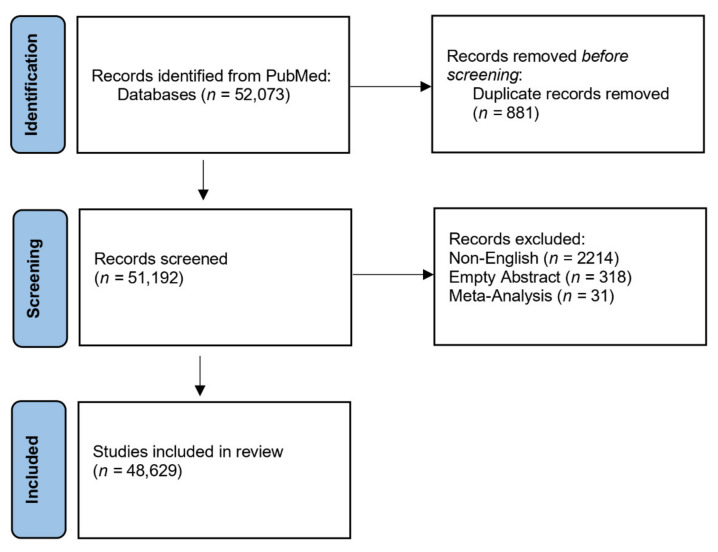
PRISMA flow diagram of the selection of studies to be included in the meta-analysis during the period ranging from 2000 (Q1) to 2021 (Q1).

**Figure 2 cancers-13-04417-f002:**
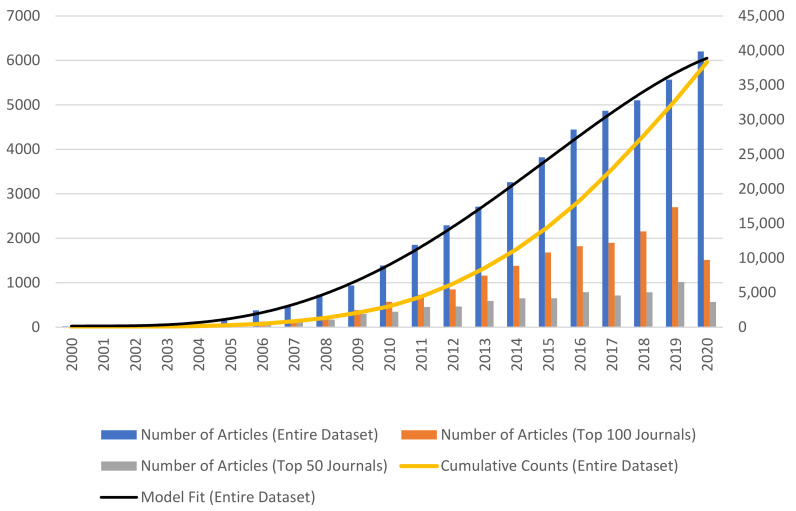
The number of articles from PubMed containing cancer nanotechnology per year from 2000 to 2021. A polynomial-based model curve $\left(y\;=-0.2439\times {\left(\mathrm{year}\right)}^{4}+8.843\times {\left(\mathrm{year}\right)}^{3}\;–\;77.138\times {\left(\mathrm{year}\right)}^{2}+221.11\times \left(\mathrm{year}\right)+5\right)$ was used to fit the global trend in “cancer nanotechnology”. The goodness-of-fit is given by $R^{2}=0.97$.

**Figure 3 cancers-13-04417-f003:**
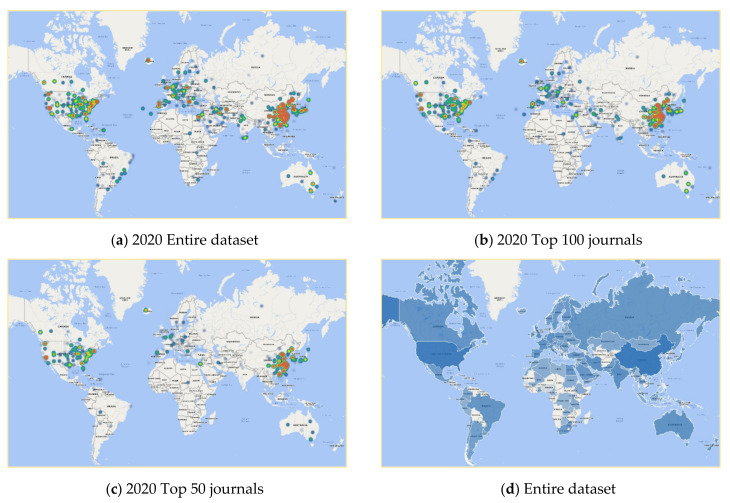
Heatmap and Geomap of the number of publications in the area of cancer nanotechnology in (**a**) the entire dataset, (**b**) the top 100 journals, (**c**) the top 50 journals, (**d**) per country for the entire dataset.

**Figure 4 cancers-13-04417-f004:**
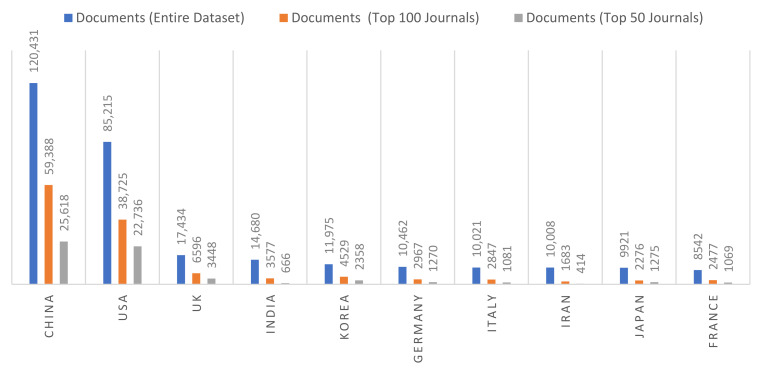
The top 10 most published records per country from 2000 to 2021.

**Figure 5 cancers-13-04417-f005:**
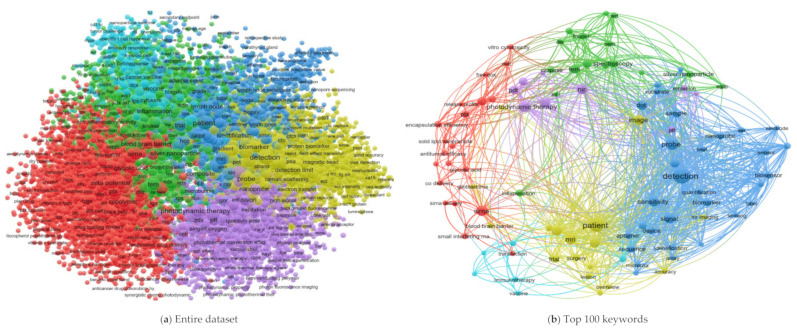
The network of co-occurring keywords in the area of cancer nanotechnology during 2000–2021 for the three scenarios in (**a**,**c**,**e**); their corresponding top 100 keywords shown in (**b**,**d**,**f**).

**Figure 6 cancers-13-04417-f006:**
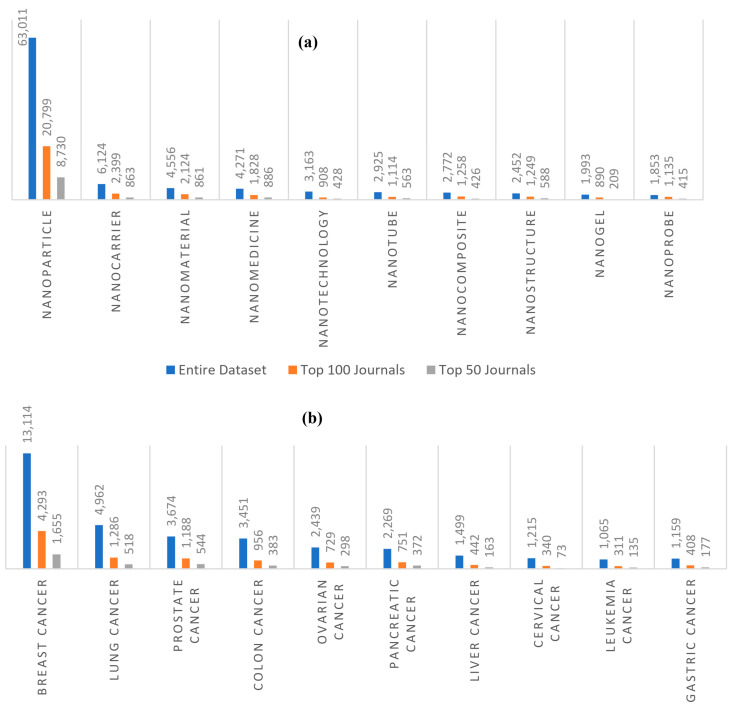
(**a**) The top 10 co-occurring nano keywords and (**b**) cancer types.

**Figure 7 cancers-13-04417-f007:**
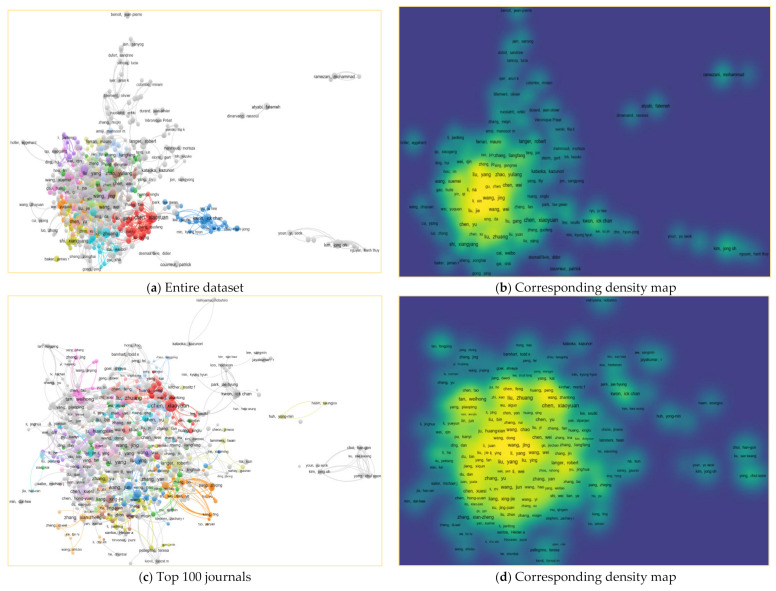
Co-authorship network and density map for cancer nanotechnology, 2000–2021, for the three scenarios in (**a**,**c**,**e**) representing co-author network for each dataset, and their corresponding density maps shown in (**b**,**d**,**f**).

**Figure 8 cancers-13-04417-f008:**
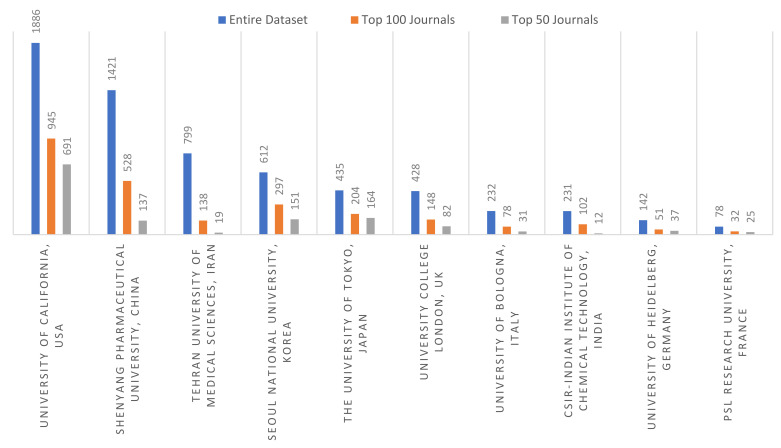
Main affiliations of authors publishing in the area of cancer nanotechnology.

**Figure 9 cancers-13-04417-f009:**
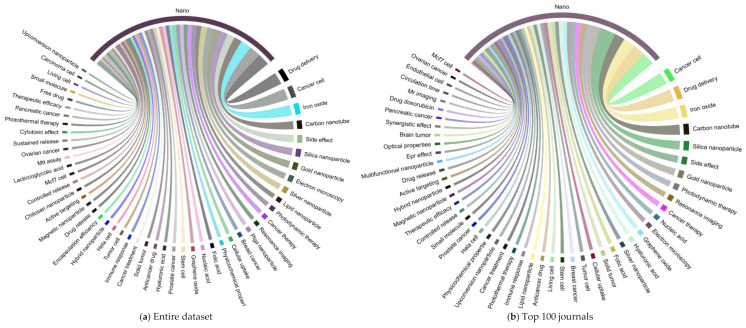
Chord diagram of research areas in nanotechnology for the three scenarios: (**a**) entire dataset, (**b**) top 100 journals, and (**c**) top 50 journals. The chord diagram was produced using the circlize package in R based on the adjacency matrix of keywords associated with nanotechnology. For better image quality, readers are encouraged to check the URL: https://1data.life/pages/publication/Cancer%20Nanotechnology.html (accessed on 20 August 2021).

**Figure 10 cancers-13-04417-f010:**
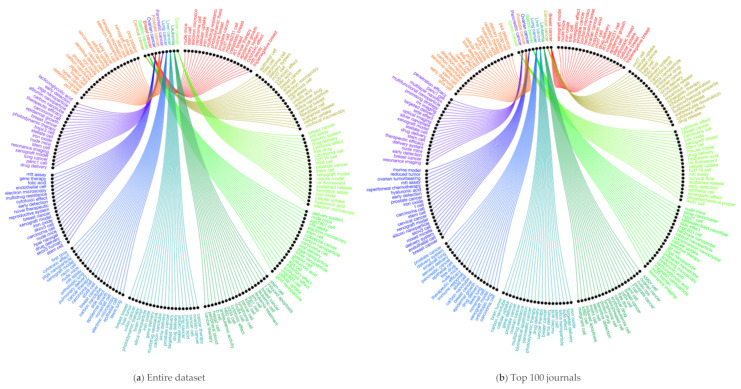
Circos plot of research areas in top 10 cancer types for the three scenarios: (**a**) entire dataset, (**b**) top 100 journals, and (**c**) top 50 journals. For better image quality, readers are encouraged to check the URL: https://1data.life/pages/publication/Cancer%20Nanotechnology.html (accessed on 20 August 2021).

**Figure 11 cancers-13-04417-f011:**
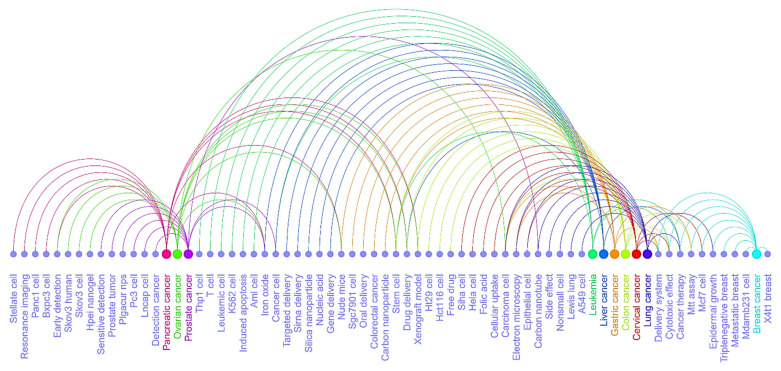
Arc diagram of overlapping research areas among the top 10 cancer types obtained from Figure 10. Large nodes are used to show cancer types and small filled circles represent the top research topics related to each cancer type linked with arc associations. The common topics can be observed by multiple incoming arcs.

**Table 1 cancers-13-04417-t001:** General information on articles related to “cancer nanotechnology” published in the period ranging from 2000 to 2021. Data is given based on the number of unique authors (from a total of 381,752 non-unique authors as each author can have more than one publication).

General Information	Entire Dataset	Top 100 Journals (5 ≤ IF ≤ 245)	Top 50 Journals (8 ≤ IF ≤ 245)
Articles	48,629	17,692	7835
Articles per author	3.88	5.853	7.981
Authors per article	0.257	0.170	0.125
Co-authors per article	5.049	4.853	6.981
Sources (journals and others)	1701	100	50
Unique authors	188,676	71,510	40,589
All authors	381,752	150,410	71,350

**Table 2 cancers-13-04417-t002:** Top 10 keywords within the entire dataset, top 100 journals, and top 50 journals in the area of cancer nanotechnology in 2000–2021.

Entire Dataset	Freq.	Top 100 Journals	Freq.	Top 50 Journals	Freq.
Therapy/Drug/Treatment	101,552	Cells	31,911	Cells	12,653
Cells	78,651	Cancer	35,379	Cancer	15,585
Cancer	77,907	Delivery	10,740	Delivery	4548
Nanoparticles	41,892	Nanoparticles	16,720	Nanoparticles	6729
Delivery	25,615	Imaging	8645	Imaging	3785
Imaging	14,855	Therapy	35,784	Therapy	6505
In Vivo	13,015	In Vivo	6086	In Vivo	2741
Targeting	10,350	Targeting	4576	Targeting	2065
MRI	8455	MRI	3926	MRI	1575

**Table 3 cancers-13-04417-t003:** Top 10 journals published in the area of cancer nanotechnology for the three scenarios with their impact factors.

Entire Dataset			Top 100 Journals (5 ≤ IF ≤ 245)			Top 50 Journals (8 ≤ IF ≤ 245)		
Journal	Freq	IF	Journal	Freq	IF	Journal	Freq	IF
International Journal of Nanomedicine	2149	4.47	ACS Applied Materials and Interfaces	1839	8.33	Biomaterials	1786	10.3
ACS Applied Materials and Interfaces	1839	8.33	Biomaterials	1786	10.27	ACS Nano	1237	13.72
Biomaterials	1790	10.27	Nanoscale	1397	6.97	Biosensors and Bioelectronics	794	9.52
Nanoscale	1397	6.97	ACS Nano	1237	13.72	Theranostics	538	8.54
ACS Nano	1237	13.71	Anal Chemistry	800	6.35	Nano Letter	445	12.28
International Journal of Pharmaceutics	1054	4.51	Biosensors and Bioelectronics	794	9.53	J Am Chem Soc	406	14.7
Scientific Reports	845	4.12	Materials Science and Engineering C	752	5.32	Nat Commun	294	11.68
Analytical Chemistry	800	6.35	Journal of Biomedical Nanotech	562	5.34	Proc Natl Acad Sci USA	227	9.55
Biosensors and Bioelectronics	794	9.52	Theranostics	538	8.54	Adv Drug Deliv Rev	219	16.66
Materials Science and Engineering C	752	5.31	Biomacromolecules	506	5.67	Nat Nanotechnology	158	33.41

## Data Availability

The datasets used in this work were publicly available at: https://1data.life/pages/publication/Cancer%20Nanotechnology.html (accessed on 20 August 2021).

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
