# Peer review of "Global Trends in Cancer Nanotechnology: A Qualitative Scientific Mapping Using Content-Based and Bibliometric Features for Machine Learning Text Classification"

_cancers, 2021, doi:10.3390/cancers13174417_

Round 1

Reviewer 1 Report

The authors have addressed the recommendations. Different sections have been introduced, so I recommend it for publication. And a different section of the paper has been rewritten, so the main ideas have been exposed clearly.

The work is very well written.

Author Response

Response: We truly appreciate your careful review and comments.

Reviewer 2 Report

Some comments sorted by appearance in the manuscript:

- The statement "cancer nanotechnology can improve a large number of scientific applications in society" has to be clarified or changed.

- Please add references to introduction as to "advancements in cancer nanotechnology..."

- Readers would highly benefit from mentioning more about bibliometrics with possibly the following:
https://ieeexplore.ieee.org/abstract/document/4349633/: "Bibliometrics and Social Network Analysis of the Nanotechnology Field" 
https://elifesciences.org/labs/ef274c83/open-knowledge-maps-a-visual-interface-to-the-world-s-scientific-knowledge: "Open Knowledge Maps" 

- Please give more details on how you "implement the qualitative method".

- Ranking journals by IF solely is not suitable if several domains are compared!
In this case journals would have to be evaluated by field-related quartiles.

- Regarding reproducibility readers may be interested in finding a link to sources of your used scripts to extract key elements and the number of citations. 
Readers may also be interested in finding more information on the evaluation regarding the quality estimation related to IF: though results look promising, did you think of using other metrics too?

- Just as an idea, readers may also benefit from, additionally to the chord diagrams, adding a dendogram of the top overlapping topics.

- Discussion and conclusion have similiar size and should be fused or conclusion used as highlight over a long discussion.

Round 2

Reviewer 2 Report

Thank you for your friendly response, authors handled all concerns.
Some further remarks:
- In regard to the description of the qualitative method, one side-note, that abbreviations should be handled with first they have been mentioned
- The focus of cancer nanotechnology has been mentioned within the abstract and should additionally be noted within the methods section, as reason why distinct field-related quartiles have not been utilized, the related challenges could be expatiated within the discussion section.

Author Response

Thank you for your friendly response, authors handled all concerns.

Response: Again we truly appreciate the constructive comments and suggestions from the reviewer that greatly improve the quality of the manuscript.

Some further remarks:
- In regard to the description of the qualitative method, one side-note, that abbreviations should be handled with first they have been mentioned

Response: This was fixed and highlighted in the manuscript.

- The focus of cancer nanotechnology has been mentioned within the abstract and should additionally be noted within the methods section, as reason why distinct field-related quartiles have not been utilized, the related challenges could be expatiated within the discussion section.

Response: This was a good idea. We changed our method and added two paragraphs in the Discussion in the manuscript. Below is the edited part in the Method section:

"Compared to all publications, we used the journal-level metric IF as a scientometric index introduced by the Institute for Scientific Information and currently published by Clarivate [27] to further divided the dataset into the top 100 journals (5 ≤ IF ≤ 245) and top 50 journals (8 ≤ IF ≤ 245) with respect to high quality/high impact publications. It is worth mentioning that there are a number of publications indexes and factors about measuring scientific activities and publication impacts including articles citations, h-index, i10-index, SJR Q indexing, etc [28, 29], but we eventually decided to utilize the journal-level metric IF which may represent the better measure as this journal-level metric is globally accepted."

and then added our previous discussion with the reviewer about IF as follows:

"When this project was started, the focus of our work has been to perform research topics specifically related to cancer nanotechnology. For this purpose, finding a perfect measure to further divide the documents based on keywords and main themes was not a straightforward task and that is the main reason we selected an internationally recognized measure, IF, to index and parse all the publication records. The advantage of using this journal-level metric was that we were able to investigate three datasets (Entire Dataset, Top 100 journals, and Top 50 Journals) and made a comparative analysis to identify how significantly high-quality/high-impact publications change the course and field of cancer nanotechnology. This was clear that the entire dataset containing almost 50,000 published documents provides all field-related information about cancer nanotechnology. Using IF≥5, we were then able to implement our data-mining techniques on papers from the top 100 journals which, roughly speaking, covers almost all journals in the field of nanotechnology. Finally, it is reasonable to say that the benefit of using IF≥8 was that it could potentially include all the top journals that disseminate manuscripts in nano-related fields. Using this strategy, we could clearly see global patterns and changes when we applied this formula to our three datasets, for instance, see Tables 2 and 3 or Figures 9 and 10.

Another important reason that we could not make use of a different journal-level metric than IF was that the journal’s aims and scopes usually cover a broad range of topics which are typically inconclusive to select a very specific field of research. We understand that it is necessary and imperative for them to cast a wide net of research topics to bring together extensive research relevant to the journal's audience as one of their main goals is to target a large, general readership. For this reason, we believed this might have been a convoluted task to identify other measures than IF to filter for different datasets. Also, we had not had a priori knowledge to do predefined filtering based on different topics or cancer types, or nano-related materials. It first necessitated to analyze the data and then see the trends in each field to comprehend whether it was required to identify any other measures or not. Now that different patterns can be observed using this measure and the techniques used in this study, it would be noteworthy to further mine the data for other types of patterns using these key research topics for future work. As previously mentioned, one limitation we may anticipate is that identifying a specific field might involve arduous labor to distinguish relevant journals and that journals cover a wide range of research topics in their scopes and aims."